# Pulsed Blue Light and Phage Therapy: A Novel Synergistic Bactericide

**DOI:** 10.3390/antibiotics14050481

**Published:** 2025-05-09

**Authors:** Amit Rimon, Jonathan Belin, Ortal Yerushalmy, Yonatan Eavri, Anatoly Shapochnikov, Shunit Coppenhagen-Glazer, Ronen Hazan, Lilach Gavish

**Affiliations:** 1Institute of Biomedical and Oral Research (IBOR), Faculty of Dental Medicine, The Hebrew University of Jerusalem, Jerusalem 9112102, Israel; amit.rimon@mail.huji.ac.il (A.R.); jonathan.belin@mail.huji.ac.il (J.B.); ortal.yerushalmy@mail.huji.ac.il (O.Y.); shunitc@ekmd.huji.ac.il (S.C.-G.); ronenh@ekmd.huji.ac.il (R.H.); 2Tzameret, The Military Track of Medicine, The Hebrew University-Hadassah Medical School, Jerusalem 9112001, Israel; 3The Israeli Phage Therapy Center (IPTC) of the Hebrew University and Hadassah Medical Center, Jerusalem 9112001, Israel; 4Institute for Research in Military Medicine (IRMM), Faculty of Medicine, The Hebrew University of Jerusalem, Jerusalem 9112001, Israel; yonatan.eavri@gmail.com; 5The Department of Medical Neurobiology, Institute for Medical Research (IMRIC), Faculty of Medicine, The Hebrew University of Jerusalem, Jerusalem 9112001, Israel; anatolys@ekmd.huji.ac.il; 6The Saul and Joyce Brandman Hub for Cardiovascular Research, Jerusalem 9112001, Israel

**Keywords:** bacteriophage, phage therapy, photobiomodulation, blue light, biofilm, *Pseudomonas*

## Abstract

**Background:** Antibiotic-resistant *Pseudomonas aeruginosa* (*P. aeruginosa*) strains are an increasing cause of morbidity and mortality. Pulsed blue light (PBL) enhances porphyrin-induced reactive oxygen species and has been clinically shown to be harmless to the skin at low doses. Bacteriophages, viruses that infect bacteria, offer a promising non-antibiotic bactericidal approach. This study investigates the potential synergism between low-dose PBL and phage therapy against *P. aeruginosa* in planktonic cultures and preformed biofilms. **Methods:** We conducted a factorial dose–response in vitro study combining *P. aeruginosa-specific* phages with PBL (457 nm, 33 kHz) on both PA14 and multidrug-resistant PATZ2 strains. After excluding direct PBL effects on phage titer or activity, we assessed effectiveness on planktonic cultures using growth curve analysis (via *growth_curve_outcomes*, a newly developed, Python-based tool available on GitHub) , CFU, and PFU. Biofilm efficacy was evaluated using CFU post-sonication, crystal violet staining, and live/dead staining with confocal microscopy. Finally, we assessed reactive oxygen species (ROS) as a potential mechanism using the nitro blue tetrazolium reduction assay. ANOVA or Kruskal–Wallis tests with post hoc Tukey or Conover–Iman tests were used for comparisons (*n* = 5 biological replicates and technical triplicates). **Results:** The bacterial growth lag phase was significantly extended for phage alone or PBL alone, with a synergistic effect of up to 144% (*p* < 0.001 for all), achieving a 9 log CFU/mL reduction at 24 h (*p* < 0.001). In preformed biofilms, synergistic combinations significantly reduced biofilm biomass and bacterial viability (% Live, median (IQR): Control 80%; Phage 40%; PBL 25%; PBL&Phage 15%, *p* < 0.001). Mechanistically, PBL triggered transient ROS in planktonic cultures, amplified by phage co-treatment, while a biphasic ROS pattern in biofilms reflected time-dependent synergy. **Conclusions:** Phage therapy combined with PBL demonstrates a synergistic bactericidal effect against *P. aeruginosa* in both planktonic cultures and biofilms. Given the strong safety profile of PBL and phages, this approach may lead to a novel, antibiotic-complementary, safe treatment modality for patients suffering from difficult-to-treat antibiotic-resistant infections and biofilm-associated infections.

## 1. Introduction

### 1.1. Antibiotic-Resistant Bacteria and Biofilms

The rise in the prevalence and spread of antibiotic-resistant bacteria is a critical health concern, directly contributing to 1.27 million deaths in 2019 alone [1]. The genetic plasticity of bacterial populations, driven by selective pressures from the overuse of antibiotics in clinical, agricultural, and environmental settings, facilitates rapid bacterial adaptation and resistance mechanisms [2]. This phenomenon significantly diminishes the effectiveness of conventional antibiotic therapies, particularly in biofilms—structured bacterial communities encased in a self-produced extracellular matrix. These microbial aggregates enhance resistance to both antimicrobial agents and host immune responses, presenting substantial challenges in clinical and industrial environments [3]. *Pseudomonas aeruginosa* (PA), a Gram-negative rod, is one of the six leading pathogens responsible for antibiotic resistance-related deaths [1], and it frequently forms biofilms, especially in burns and other wounds [4]. Therefore, novel treatments, strategies, or combinations of approaches are urgently needed to combat this pathogen. In this study, we explore two such approaches and demonstrate their synergistic effects against PA.

### 1.2. Phage Therapy

A growing and revitalized approach to combat resistant bacteria is the use of bacteriophages (phages) [5], viruses that specifically target bacteria without harming eukaryotic cells. Phages are the most abundant biological entities on Earth, with an estimated 10^31^ particles worldwide [5]. Unlike antibiotics, which can affect a broad range of bacteria, phages target specific bacterial strains while largely sparing other microbiota [5]. One key advantage of phages as therapeutic agents is their ability to penetrate biofilm structures [6,7]. Furthermore, bacterial resistance to phages can be managed through novel phage isolation, phage engineering, or naturally through evolutionary changes [8]. Phage therapy has shown effectiveness against several antibiotic-resistant pathogens including *Pseudomonas aeruginosa*, *Staphylococcus aureus*, and *Cutibacterium acnes* [9,10,11], particularly in skin-related infections. Additionally, phage therapy has been successfully used in compassionate cases and case series [9,12] including by our team in the Israeli Phage Therapy Center (IPTC) to treat non-healing infections caused by antibiotic-resistant bacteria [13].

### 1.3. Pulsed Blue Light (PBL)

Another promising approach to combat resistant bacteria is antimicrobial blue light, which has been shown to be effective against both Gram-positive and Gram-negative bacteria, particularly antibiotic-resistant *P. aeruginosa* strains [14]. Blue light photons are absorbed by bacterial chromophores, such as porphyrins (notably coproporphyrin in *P. aeruginosa* [15,16]), leading to the generation of reactive oxygen species (ROS) that induce oxidative damage including to the bacterial membranes [17]. In addition to ROS-mediated effects, blue light has also been reported to directly depolarize the membrane potential and disrupt membrane integrity [17,18]. Clinical studies have demonstrated that blue light at low doses is harmless to the skin [19]. Pulsed blue light (PBL) has been found to have greater bactericidal efficacy compared to continuous blue light, requiring much lower power intensity and energy doses in planktonic cultures [20]. Additionally, PBL has been shown to partially disrupt and disassemble biofilm structures [21].

### 1.4. Study Rationale and Objective

The rise in antibiotic-resistant bacteria and the limited availability of new antibiotics [5] have created an urgent need for alternative treatments. In response, we explored the combination of pulsed blue light (PBL) and phage therapy as a novel non-antibiotic strategy. These two approaches target bacteria through distinct, complementary mechanisms: PBL generates reactive oxygen species (ROS) that disrupt bacterial membranes and biofilm structures [17,20], while phages specifically infect and lyse bacteria, including those within biofilms [6]. Given that both PBL and phages are effective against antibiotic-resistant pathogens, combining them may provide a synergistic effect that enhances bacterial eradication without contributing to the growing problem of antibiotic resistance.

This study aimed to assess the efficacy of combining PBL and phage therapy on both planktonic and biofilm cultures of *Pseudomonas aeruginosa*, a model organism frequently associated with antibiotic-resistant infections. By demonstrating the potential for synergy between these two non-antibiotic treatments, we hope to provide a new approach to combating resistant bacterial infections, particularly in biofilm-associated cases.

## 2. Results

### 2.1. Effect of PBL on Phage Number and Activity

To rule out the direct effect of 450 nm PBL on phages, we first exposed phages to the extended duration of PBL and quantified their titer and bactericidal activity. This was important for determining the timing of PBL administration in relation to phage administration. As expected, no reduction in the titer of either PAShipCat1 or PASA16 was observed after up to 3 h of PBL irradiation (*p* > 0.5 for both phages, by 1-way ANOVA). Furthermore, the bactericidal activity of the irradiated phages against planktonic cultures or preformed biofilms was not impaired, as assessed by CFU (*p* < 0.001 for all PBL doses) (Figure 1 and Appendix A).

### 2.2. Timing of Administration and Dose/Concentration Response

After confirming that 457 nm 33 kHz PBL irradiation does not directly affect phage activity, we proceeded to evaluate the combination of PBL and phage treatments. Initially, we tested the combination using the lytic phage PASA16; however, due to its complete bactericidal effect (See Section 4 below), we isolated a novel, less effective phage with a lysogenic lifecycle, PAShipCat1, to assess the potential synergism between phage and PBL. The timing of phage addition to planktonic *P. aeruginosa* in relation to PBL was evaluated using PAShipCat1 (10^6^ PFU) at 10, 90, and 150 min post-PBL treatment (8.5 J/cm^2^). While PBL and phage treatments alone reduced the area under the curve (AUC) at the 20 h post-PBL treatment compared to the controls by 28% and 51%, respectively, their combination resulted in a 73% reduction, with the most effective timing being the 10 min post-PBL treatment (Figure 2A and Appendix A).

Using the above protocol, we tested the effect of different phage concentrations combined with various PBL power intensities. Phage treatment alone significantly extended the lag phase, as did PBL alone (*p* < 0.001 for all concentrations and power intensities). When PBL and phages were combined, an additive effect was observed at 10^2^/10^4^ PFU/mL, while a synergistic effect of up to 144% was noted at higher phage concentrations (Figure 2B). This combination reduced CFU by 9 log in PA14 and also produced significant results when applied to the multidrug-resistant (MDR) strain PATZ2, although to a lesser extent (Figure 3 and Appendix A).

### 2.3. Bacterial Viability and Biomass

Next, we assessed the effect of the PBL–phage combination on preformed biofilms. First, we evaluated the optimal timing for phage addition relative to the PBL doses. PASA16 phage was used, as it has greater anti-biofilm activity than PAShipcat1. Using crystal violet staining, we found that both PBL and phage treatments reduced biofilm biomass, with the most effective protocol being the addition of phage immediately before PBL regardless of the dose (*p* < 0.001 across doses, presence of phage, and timing of phage addition, by 3-way ANOVA) (Figure 4A and Appendix A). Based on these results, we chose to add phage immediately before a 26 J/cm^2^ PBL treatment for subsequent experiments.

CFU analysis revealed that this combination achieved a synergistic 3-log reduction (Figure 4B). Using live/dead staining, we found that all treatments significantly decreased bacterial viability (*p* < 0.001 vs. controls). However, while phage alone reduced bacterial viability by 50% compared to controls, the addition of PBL further reduced bacterial viability to 19% (*p* = 0.012 for PBL&Phage vs. Phage alone) (Figure 4C,D and Appendix A).

### 2.4. Reactive Oxidant Species

Finally, to determine the mechanism underlying the synergistic effect, we measured the level of intracellular reactive oxygen species. In planktonic culture, PBL induced an immediate increase in ROS that was significantly amplified to 169% of the controls with the addition of phage (*p* < 0.001). In the biofilm, no change in ROS levels was observed with PBL or phage alone. However, ROS levels following the addition of the combination exhibited a biphasic modulation over time, with a significant decrease at 1 h after the addition of phage but a significant increase at 3 h (65% and 154% of controls; *p* = 0.012 and *p* = 0.002, respectively, Figure 5 and Appendix A).

## 3. Discussion

Antibiotic-resistant bacteria are a major health concern. In this study, we explored a combination of two non-antibiotic bactericidal techniques: pulsed blue light (PBL) and phage therapy. While combining antibiotics with phages has shown promise, the overuse of antibiotics is driving resistance, making non-antibiotic options increasingly important [23].

PBL and phage therapy are innovative because both therapies act independently of antibiotics. PBL damages bacterial cell structures and increases susceptibility to phages, which specifically target bacteria for lysis. Additionally, whereas antibiotics can harm beneficial bacteria and promote resistance [23,24], PBL can be applied locally, minimizing systemic side effects, and phages are strain-specific. Thus, PBL and phage therapy offer a focused, non-antibiotic alternative that could address antibiotic resistance while maintaining high bactericidal efficacy.

The results of this study show that both planktonic and biofilm cultures of *P. aeruginosa* are highly susceptible to combined treatment of phage therapy and pulsed blue light. The synergistic effect of this combination led to a 9-log reduction in planktonic cultures and a 40% reduction in preformed biofilms. The most effective bactericidal results were observed at a dose of 26 J/cm^2^, with 10^8^ PFU/mL phage added within 10 min of PBL administration. This combination of PA-targeted phage therapy and low-dose, no-risk pulsed blue light presents a promising strategy for treating superficial wound infections, such as burns, acne, diabetic foot ulcers, and cellulitis. Additionally, it could enhance biofilm management, a growing escalating challenge in the context of antibiotic resistance [1].

### 3.1. Phages and Antibiotics

Phage therapy has a well-established safety profile, supported by numerous clinical trials and compassionate use cases, and has shown efficacy with minimal adverse effects in treating *Pseudomonas aeruginosa* infections [25]. Currently, phage therapy is usually combined with antibiotic treatment [26]. Lin et al. reported that the effectiveness of the combination depends on the *P. aeruginosa* strain and the antibiotic used. For example, phage PEV20 achieved complete inhibition in planktonic cultures when combined with ciprofloxacin but not with amikacin or colistin [27]. However, Knezevic et al. reported that the combination with ciprofloxacin led to only a 3.3-log reduction in PA [28]. Similarly, in this study, we observed complete inhibition of planktonic PA14 but partial inhibition of MDR PATZ2. Interestingly, antibiotics alone were ineffective in reducing *Pseudomonas* biofilm biomass [29,30]. However, Chaudhry et al. demonstrated that combining 10 µg/mL ciprofloxacin with phages resulted in a 47–74% reduction in biofilm biomass from a clinical cystic fibrosis *P. aeruginosa* strain, which was higher than the 40% synergistic effect observed in the current study [29]. Although increasing the PBL dose did not further reduce biomass (Figure 4A), increasing the power intensity may enhance eradication efficacy (Figure 3B), which will be the focus of future studies.

### 3.2. Blue Light

While blue light may be harmful to the eyes, skin exposure has been clinically determined to be harmless [19]. Clinical studies demonstrate that low doses of PBL is harmless to the skin, and LED patches with similar specifications are FDA-approved and available for consumer use in the USA [19]. Pulsed blue light in similar frequencies and doses is sold commercially as LED patches (in combination with red LEDs) for personal health-related use in the USA and has FDA approval (https://www.carewear.net/). In a clinical study, healthy volunteers were exposed to a daily dose of 100 J/cm^2^ of 420 nm blue light for 5 consecutive days. No indication of DNA damage, inflammation, or apoptosis was detected in their skin biopsies, although transient keratinocyte melanogenesis and vacuolization were observed [31]. The bactericidal effect of continuous blue light has been extensively studied, with special attention given to the trade-off between power and dose to avoid a photothermal effect: Rupel et al. used 455 nm blue light with an irradiance of 300 mW/cm^2^ and a dose of 60 or 120 J/cm^2^ to irradiated planktonic cultures of *P. aeruginosa* achieving inhibition for 24 h [32]. In the current study, we achieved a post-infection 9-log CFU/mL reduction using a similar wavelength, 7.2 mW/cm^2^ irradiance, and a dose of 13 J/cm^2^. Pulsed light allows for higher peak power densities compared to those used in continuous wave treatments, without exposing the skin to excessive tissue heating [20]. Indeed, PBL in very low average irradiances and doses has been shown to be superior to continuous wave light. Bumah et al. demonstrated that PBL, using multiple 30 min sessions of 450 nm PBL at doses of 7.6 J/cm^2^ or lower, was more effective than continuous wave light at similar doses. They reported a 7-log CFU/mL reduction in planktonic cultures and 64% and 53% suppression of biofilms in *C. acnes* and MRSA, respectively [21]. In comparison, we achieved a 9-log reduction in planktonic *P. aeruginosa* culture and 40% suppression of pre-formed *P. aeruginosa* biofilms, highlighting the variability of the response between different bacterial species.

### 3.3. Practical Considerations

We found that the most effective timing for treatment was administering PBL immediately after phage administration in biofilm cultures (Figure 3 and Figure 4). The synergistic effect remained significant even when phage was administered 24 h before PBL, as would be the case with intravenous or oral phage delivery [33]. Our study utilized pulsed blue light with clinically applicable power density and pulse frequency [21,34], making our findings relevant to potential clinical applications [35]. Notably, one of the two phages used, PASA16, has previously been involved in various compassionate clinical treatments [9].

### 3.4. Possible Mechanisms

The proposed synergy between phage therapy and pulsed blue light (PBL) likely involves multiple mechanisms, including lysogen activation, which similar to ultraviolet light [36] and overwhelms the cell’s DNA repair mechanisms, as both treatments cause DNA damage [37], and ROS induction, which we tested as part of this study. In planktonic cultures, we observed that PBL immediately induces ROS, which is known to activate bacterial antioxidant defenses [38], and has also been shown to cause membrane perturbations (Enwemeka et al., 2017 and 2021 [17,18]). The increased ROS levels may facilitate phage entry, intensifying oxidative stress. However, biofilms present a more complex scenario. Repeated PBL exposure likely triggered early antioxidant responses in biofilm-residing cells, buffering ROS levels and preventing detectable accumulation. Meanwhile, phage infection was delayed due to the expected interference of virion diffusion via the extracellular polymeric substance [39]. In the combined treatment, membrane damage by PBL facilitated phage entry and likely accelerated the infection rate, prompting a stronger antioxidant surge that suppressed ROS at 1 h. By 3 h, progressing phage replication and cell lysis may have released secondary ROS [40]. Future experiments should aim to validate this hypothesis by measuring antioxidant gene expression, intracellular iron levels, and membrane integrity at key time points.

### 3.5. Limitations

This study was limited by its focus on only two phages (PASA16 and PAShipCat1) and two *P. aeruginosa* strains (PA14 and PATZ2), which restricts the generalizability of our findings. The treatment duration of 30 min may also be considered lengthy in practical applications. However, we used very low-dose PBL, which was effective and presented the opportunity to explore higher doses that could reduce treatment time, all while remaining harmless to the skin.

## 4. Materials and Methods

### 4.1. Study Design

In this study, we employed a factorial dose–response design involving combinations of phage concentrations and PBL power densities to test their bactericidal effect against *P*. *aeruginosa* planktonic cultures during the logarithmic phase and on pre-formed biofilms (Figure 6A–E).

### 4.2. Bacteria and Growth Conditions

#### 4.2.1. Bacterial Strains

The bacteria used in this study were the reference *Pseudomonas aeruginosa* strain PA14 [41], obtained from our laboratory stock collection, and the multidrug-resistant strain MDR PATZ2, which was isolated from a clinical phage therapy case in our institution [9]. MDR PATZ2 was determined to be resistant to 2 μg/mL Ceftazidime (Panpharma Z.I, Luitre, France). Antibiotic resistance was assessed by the Hadassah Medical Center Clinical Microbiology Laboratory.

#### 4.2.2. Planktonic Bacteria

Bacteria were grown from a −80 °C stock overnight in Luria Bertani (LB) medium at 37 °C under continuous agitation at 60 RPM. The cultures were then diluted 1:1000 and re-grown in LB. Bacteria at the mid-logarithmic phase (OD_600_ ≅ 0.5) were used for the experiments.

#### 4.2.3. Biofilm

Bacteria (200 μL/well, 10^8^ CFU/mL) were grown overnight and then transferred into the inner wells of a 96-well plate made from virgin polypropylene, where they were incubated for 48 h. The outer wells were filled with double-distilled water to prevent uneven evaporation [42].

### 4.3. Phage Strains

In this study, we used two phage strains specific for PA: one lytic and the other lysogenic (PASA16 and PAShipCat1, respectively).

The phage PASA16 was previously isolated and characterized in our lab [43] and has been used in various clinical compassionate treatments [9]. As described previously [43], PAShipCat1 is a novel lysogenic phage isolated and characterized in this work [44].

These phages were deliberately chosen due to their moderate effects, which allowed us to observe potential synergistic effects with the PBL, despite their varying efficacy.

### 4.4. PBL

The light source was designed in-house to provide uniform irradiation over a 96-well plate inside a humid incubator (Figure 7). The irradiating surface consisted of 457 nm blue LED strips (Shenzhen Dengsum Optoelectronics Co., Ltd., Shenzhen, China) with high-lumen 2835 SMD chips (Epistar Corp., Hsinchu, Taiwan) arranged at a density of 120 LEDs/m. The device was powered by an LED driver (LCM-40, Mean Well Enterprises Co., Ltd., New Taipei City, Taiwan). Pulses, with a frequency of 33 kHz and a 50% duty cycle, were generated using an electronic circuit based on a 555 pulse generator module. Power density was measured at the plane of the cells using a LaserMate power meter (Coherent, Auburn group, Coherent-Europe, Utrecht, The Netherlands). The PBL frequency and initial power density were based on published experiments by Enwemeka et al. in collaboration with CareWear company (Reno, NV, USA) [17,20,21].

### 4.5. Quantification Methods

#### 4.5.1. CFU

Samples were serially diluted 10-fold onto LB agar plates and incubated overnight at 37 °C. The number of colonies was counted, and the concentration of CFU was calculated.

#### 4.5.2. PFU

The lysate was diluted 10-fold in LB broth and spotted onto a PA14 bacterial lawn grown on soft 0.6% agar, then incubated overnight at 37 °C. Plaques were observed and counted, and the concentration of PFU was calculated.

#### 4.5.3. Growth Curves

Lytic activity was assessed by inoculating logarithmic *P. aeruginosa* (10 CFU/mL) with the experimental samples and measuring turbidity with a plate reader (Synergy, Gen5; BioTek, Winooski, VT, USA) at OD_600nm_ at 37 °C, with 5- s shaking every 20 min for 24 h. The data were smoothed using a 4-point rolling average. The threshold for growth curve detection was based on the measurements of the control group. The lag time is defined as the time (in hours) required for the OD to exceed the detection threshold, and the area under the curve at 20 h (AUC 20 h), which represents the growth potential (Figure 5B), was automatically extracted using the Python code *growth_curve_outcomes version 1.0* developed for this study. This code can be easily applied to plate reader growth curve data and is freely accessible on GitHub (https://github.com/bioimagehuji/growth_curve_outcomes/) (accessed on 5 May 2025).

#### 4.5.4. Biofilm CFU

Biofilms were washed with a saline solution and then mechanically removed using a sterile tip. The contents of each well were transferred into a 1.5 mL tube, and the tubes were placed in a SONOREX sonication water bath (Bandelin, Berlin, Germany) for 5 min to effectively disrupt the biofilm. Next, 5 µL of 10-fold serial dilutions were plated on LB agar plates and incubated for 24 h, and colonies were counted.

#### 4.5.5. Biofilm Biomass Detection

Biofilm biomass was measured using crystal violet staining [45] (Glentham Life Sciences, Corsham, UK). The liquid containing unattached bacteria was removed by pipetting, and the wells were rinsed with 100 μL of 0.9% NaCl and dried. A solution of 0.1% crystal violet was added to the wells (125 μL per well) and incubated for 15 min at room temperature. The dye was removed, and the wells were rinsed twice in a distilled water bath. The plate was dried at 37 °C for 1 h, and color was extracted with 30% acetic acid (100 μL per well). The purple-colored solution was transferred into a new plate, and OD_550nm_ was determined using The Synergy H1 plate reader (Agilent Technologies, Winooski, VE, USA) [45].

#### 4.5.6. Live-Dead Stain

Bacterial viability was measured using the Live/Dead Cell Viability Kit (Invitrogen, Waltham, MA, USA) according to the manufacturer’s instructions. Propidium iodide stained the dead cells red, while SYTO9 stained the live cells green (measured at 630 nm and 520 nm, respectively). The fluorescence emissions of the samples were detected using a Zeiss LSM410 confocal laser microscope (Carl Zeiss, Oberkochen, Germany) at ×4 and ×20 magnification. The maximal intensity of the Z-stack for each color was determined from multiple horizontal optical sections (17 sections at 12.5 μm intervals for ×4 and 9 sections at 10 μm intervals for ×20). Three separate fields per sample were measured using ImageJ, version 1.54 and the percentage of live bacteria was calculated as the proportion of the total bacteria following Robertson et al. [22].

#### 4.5.7. ROS Determination

Intracellular ROS levels were assessed using the nitro blue tetrazolium (NBT) assay (Cayman Chemical, Ann Arbor, MI, USA), which measures the reduction in NBT to formazan. At 10 min, 1 h, and 3 h posttreatment, 10 μL of NBT solution (2 mg/mL, freshly prepared by 1:10 dilution of a 20 mg/mL stock in DMSO) was added directly to each sample well. Absorbance at 560 nm was recorded using the Synergy H1 plate reader [46].

#### 4.5.8. Statistics

Data are presented as mean ± SD or median (interquartile range) as appropriate. The results represent the mean of five independent experiments, each with triplicate measurements (unless otherwise stated). ANOVA or Kruskal–Wallis was used to compare the groups with Tukey or Conover–Iman post hoc tests, as appropriate. A *p*-value < 0.05 was considered statistically significant. Statistical analyses were performed using SYSTAT, version 13 (Systat Software, Chicago, IL, USA).

## 5. Conclusions

We present a novel bactericidal synergistic combination of phage therapy and low-dose pulsed blue light, demonstrating effectiveness against both planktonic and preformed biofilms of *Pseudomonas aeruginosa* and proposing a mechanistic explanation for the synergistic effect based on the initial stimulation of ROS generation. Future research should include a broader range of bacterial strains, additional phages, and a comprehensive exploration of the underlying mechanisms. Regarding safety, although both phage therapy and low-dose PBL each have a high safety profile and were applied to humans, biocompatibility studies of the combined treatment in wound models are warranted. Finally, clinical trials should assess the clinical implications of the treatment proposed here for patients suffering from antibiotic-resistant superficial infections.

## Figures and Tables

**Figure 1 antibiotics-14-00481-f001:**
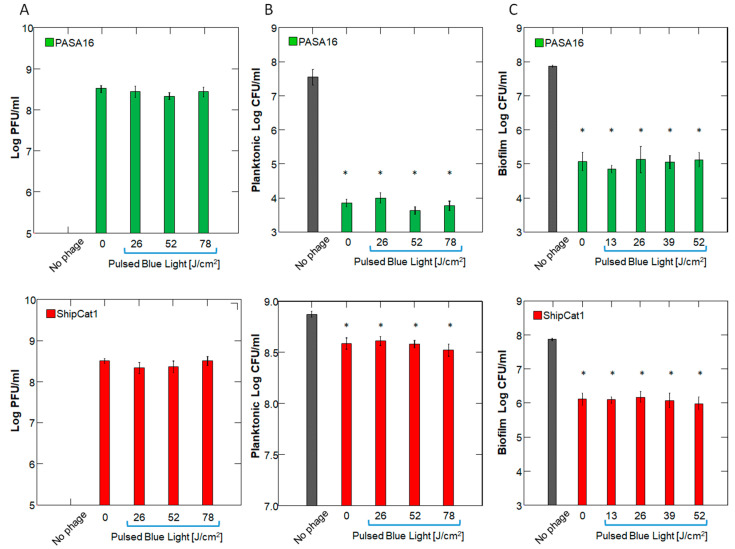
**Effect of Direct Pulsed Blue Light (PBL) on Phage Stability.** Phages PASA16 ( upper row in green) and ShipCat1 (lower row in red) a were pre-exposed to PBL at 7.2 mW/cm^2^ for 0, 1, 2, or 3 h (0–78 J/cm^2^). PA14 bacteria, either in planktonic cultures or in preformed biofilms, were then incubated with irradiated phages for 24 h. Bar graphs represent mean ± SEM for (**A**) viral titer (PFU/mL), (**B**) planktonic bacterial count, and (**C**) biofilm bacterial count post-sonication (CFU/mL). *n* = 5 biological repeats per dose (average of triplicates for each). Note that while both phages significantly reduced bacterial count, PBL did not affect viral count or bactericidal efficacy. * *p* < 0.001 compared to no phages, by ANOVA with Tukey’s post hoc.

**Figure 2 antibiotics-14-00481-f002:**
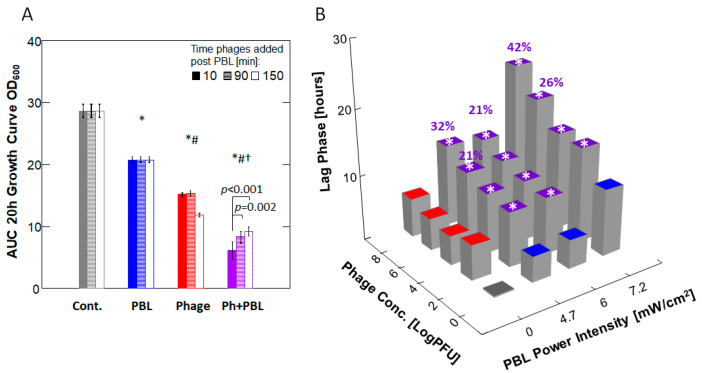
**PBL and Phage Synergism Depend on Timing and Dose**. (**A**) Bar graph showing the bactericidal effect in response to 8.5 J/cm^2^ PBL, followed by the addition of 10^6^ PFU Phage at 10, 90, or 150 min later. Bars represent the mean ± SD area under the growth curve (AUC) at 20 h; *n* = 4 (each data point is the average of triplicate technical repeats). The addition of phage at 10 min was the most effective. Statistical significance is indicated by *, #, and †, where *p* < 0.001 compared to control, PBL, and phage alone, respectively, as determined by a 2-way ANOVA with Tukey’s post hoc test. (**B**) Three-dimensional bar graph showing the bactericidal effect in response to various dose combinations of PBL followed by phage addition. Bars represent the mean lag time (in hours) compared to control, with *n* = 5 (each data point is the average of triplicate technical repeats). All comparisons were statistically significant (*p* < 0.001), as determined by a 2-way ANOVA with Tukey’s post hoc test. Additive effect: ±10% of expected; % synergistic effect: >20% over expected.

**Figure 3 antibiotics-14-00481-f003:**
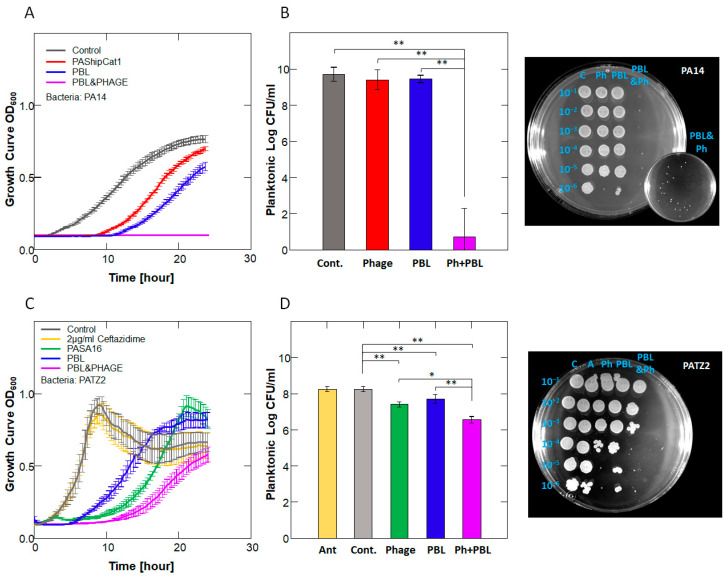
**Differential Effects of PBL and Phage on *P. aeruginosa* Strains.** *P. aeruginosa* strains were treated with or without 30 min of PBL (7.2 mW/cm^2^) followed by phage (10^8^ PFU/mL). Growth curves and CFU results are shown. (**A**,**B**) PA14 with PAShipCat1 + PBM: 9-log reduction. (**C**,**D**) Antibiotic-resistant PATZ2 with PASA16 + PBL: 1.5-log reduction. *n* = 5; representative spot titer plates for PA14 and PATZ2 (right) show serial dilutions (10⁻^1^ to 10⁻⁶) after treatment with control, phage alone, PBL alone, or PBL plus phage. The PBL + Ph group shows the most pronounced reduction in viable colonies. Inset in PA14 plate shows rare surviving colonies from the combination treatment. * *p* < 0.05; ** *p* < 0.001. Bars represent mean ± SD.

**Figure 4 antibiotics-14-00481-f004:**
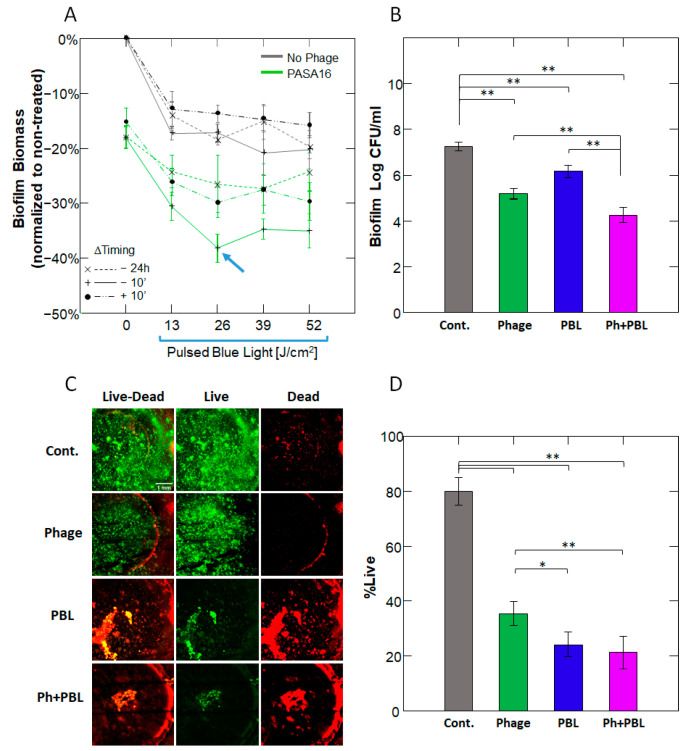
**Biofilm Viability and Biomass Are Effectively Reduced Following PBL and Phage Treatment.** (**A**) Percent reduction in biofilm biomass, normalized to non-treated controls, in response to different doses of PBL (0–52 J/cm^2^) with phage added either 10 min before PBL or 10 min or 24 h after PBL, as determined by crystal violet staining. Lines were added for visualization. The most effective combination was a PBL dose of 26 J/cm^2^ immediately after phage administration (blue arrow). Data points represent mean ± SEM. (**B**) CFU levels correspond to the most effective combination. ** *p* < 0.001, by 2-way ANOVA with Tukey’s post hoc test. (**C**,**D**) Representative live-dead fluorescently stained biofilm samples from each experimental group, visualized at 4× magnification. The percent of live bacteria was determined according to Robertson et al. [22]. Maximal intensity projections of the Z-stack, constructed by confocal microscopy, were extracted from 17 sections at 12.5 μm intervals. Bars represent mean ± SD; *n* = 5; * *p* < 0.05; ** *p* < 0.001.

**Figure 5 antibiotics-14-00481-f005:**
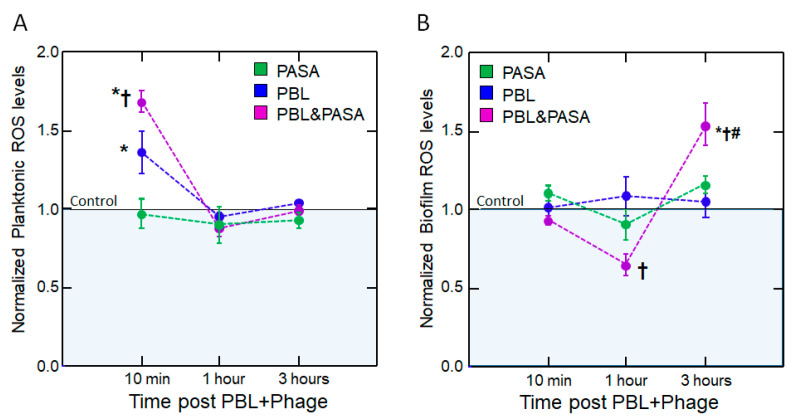
**Reactive oxygen species (ROS) levels are altered following PBL and phage treatment.** ROS levels (normalized to untreated controls; horizontal line = 1) were measured at 10 min, 1 h, and 3 h after pulsed blue light (PBL) treatment, with or without subsequent phage addition, in *P. aeruginosa* cultures. (**A**) In planktonic cultures, PBL induced an immediate increase in ROS, further enhanced by phage. (**B**) In biofilms, combined PBL and phage treatment triggered a biphasic ROS response. Points represent mean ± SEM; *n* = 5 (average of triplicates); *p* < 0.05 * vs. control, † vs. PBL, and # vs. PASA by ANOVA + Tukey.

**Figure 6 antibiotics-14-00481-f006:**
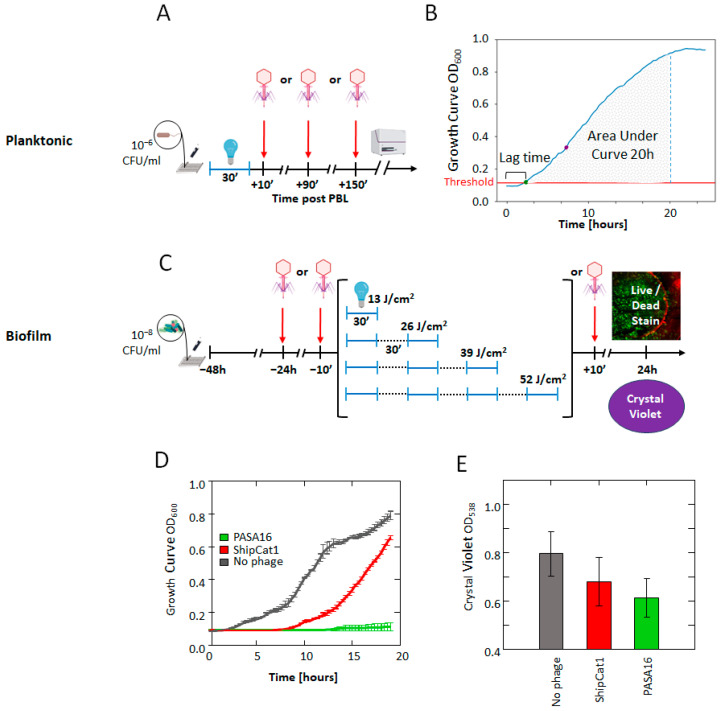
**Study Design.** (**A**) Phage was added to planktonic *P. aeruginosa* (PA14) at 10, 90, or 150 min after 30 min pulsed blue light (PBL) treatment. The bacteria were transferred to a spectrophotometer for 24 h to generate growth curves. (**B**) Measures of bactericidal effect were extracted from the growth curves, including lag time (the duration until reaching the threshold) and area under the curve at 20 h (AUC 20). (**C**) Biofilms of PA14 were formed, and phage was added 24 h and 10 min before PBL or 10 min after. PBL consisted of multiple 30 min irradiations with 30 min intervals. Crystal violet or live/dead staining was used to evaluate biofilm biomass and bacterial viability. (**D**) Phage selection for experiments with planktonic bacteria: Growth curve analysis revealed that PASA16 completely eradicated PA14, while PAShipCat1 exhibited moderate effectiveness, allowing for further investigation of its synergistic potential with PBL. (**E**) Phage selection for biofilm experiments: Crystal violet staining showed that PAShipCat1 reduced biofilm biomass with limited efficacy, whereas PASA16 demonstrated moderate efficacy and was selected for subsequent biofilm analysis. This figure was created with Biorender.com.

**Figure 7 antibiotics-14-00481-f007:**
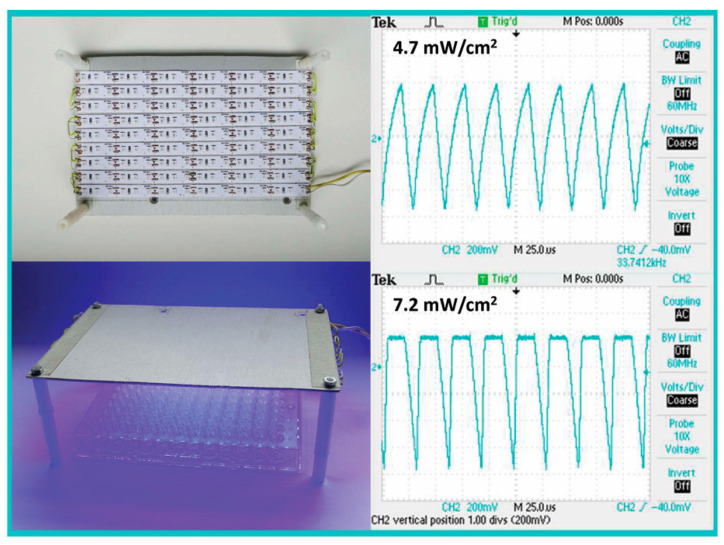
**Pulsed Blue Light Source.** Photograph of the pulsed blue light source used in this study, designed to uniformly irradiate a 96-well plate at a distance of 5 cm within a humid incubator. Stripes of 460 nm LEDs (A total of 153 LEDs) were attached to an aluminum surface and electrically interconnected. Power was supplied through an LED driver with a 555 pulse generator module. Waveform and frequency were measured using an oscilloscope connected to a light detector. Pulses were generated at a frequency of 33 kHz with a 50% duty cycle, and their waveforms were triangular and rectangular, corresponding to intensities of 4.7 and 7.2 mW/cm^2^, respectively.

## Data Availability

Phage genomes were deposited in the https://www.ncbi.nlm.nih.gov/genbank/ (accessed on 5 May 2025), and the accession numbers are MT933737.1 (PASA16) and PP067092 (PAShipCat1). The datasets generated during and/or analyzed during the current study are available from the corresponding author upon reasonable request. The *growth_curve_outcomes* Python tool version 1.0 can be accessed in Git Hub (https://github.com/bioimagehuji/growth_curve_outcomes/).

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
