# Peer review of "Pulsed Blue Light and Phage Therapy: A Novel Synergistic Bactericide"

_antibiotics, 2025, doi:10.3390/antibiotics14050481_

Round 1

Reviewer 1 Report

Comments and Suggestions for Authors

In this manuscript, Amit Rimon et al. present a study demonstrating that low-dose pulsed blue light (PBL) and phage therapy exhibit a synergistic bactericidal effect against Pseudomonas aeruginosa in both planktonic cultures and biofilms. The efficacy of this combined therapy was validated through bacterial growth inhibition assays, biofilm reduction assessments, and live/dead staining, confirming its synergistic bactericidal and antibiofilm properties. Overall, this study presents a novel approach to combating antibiotic-resistant bacterial infections. However, several key points need to be addressed before the manuscript can be considered for publication in Antibiotics. Below are my specific comments and recommendations for improvement.

  1. The innovation of this study must be emphasized in the introduction. Currently, the authors focus on background information for each therapy rather than clearly presenting the key motivation behind the research. The authors should explicitly outline what differentiates this approach from existing antibacterial strategies and how it advances current knowledge.
  2. The antibacterial mechanism of PBL remains unclear in the manuscript. Since PBL is known to induce reactive oxygen species (ROS), which contribute to bacterial inactivation, a ROS generation assay should be conducted to confirm its role in this study, which is a common method used in evaluating ROS-induced antibacterial effects. The authors should provide experimental evidence or reference relevant literature to support the proposed antibacterial mechanism.
  1. The inclusion of agar plate images with bacterial colonies under different treatment conditions is strongly encouraged. This would provide a straightforward visual representation of bacterial inhibition.
  2. Since antibacterial treatments are often applied in wound healing and other tissue environments, it is critical to assess the potential cytotoxic effects of PBL and phage therapy on normal human cells. The authors should consider conducting a biocompatibility evaluation, such as cell viability assays, to ensure that the proposed therapy does not cause harm to host tissues.
  3. While the manuscript includes a sufficient number of references, many of them are outdated. The authors should update their citations to include more recent and relevant studies to strengthen the scientific context. Additionally, reference formatting should be made consistent. Some references include DOIs, while others do not.
Comments on the Quality of English Language

Good

Reviewer 2 Report

Comments and Suggestions for Authors

This manuscript described a bactericidal therapy that combined pulsed blue light and phage to Pseudomonas aeruginosa. By combining the clinically safe and effective pulsed blue light and bactericidal phage for P. aeruginosa, the authors found that the growth of planktonic and preformed biofilms was significantly inhibited than using either PBL or phage alone. The authors also investigated the concentration and timing effects on the therapeutic results of phages along with PBL. Though the combined therapy improved the bactericidal results, here are a few questions that need to be addressed before recommended for publication

  1. The study design in section 2.1 is unclear. In Figure 1A, the figure caption said only PAShipCat1 phage was added to the planktonic culture, and in Figure 1C, the caption mentioned only PASA16 was added to the preformed biofilm. However, in Figure 1D and 1E, the phage selection should indicate that both phages were added to planktonic and preformed biofilms to get the results. The authors should clarify whether only one or both phages were added to two different cultures. I am assuming that Figure 1A and 1C described the study design after phage selection, but the current description and graph assembly could lead to huge misunderstandings to the audience.
  2. The rationale for combining PBL and phage should be mentioned in the introduction part. The current structure of the manuscript made an impression the rationale was figured after the authors found that two random things could be combined and showed improved effectiveness.
  3. I agree that the two therapies have improved bactericidal effects when combined. However, the evidence for the synergistic effect is not convincing enough for me. The PBL or phage therapy alone has good efficacy in inhibiting bacterial growth, and phage therapy showed better results than PBL alone. However, even if the two therapies are orthogonal with each other, they could still show improved therapeutic effects. They are not necessarily synergistic when combined and showing better bactericidal results. The authors demonstrated that the blue light irradiation did not affect phage growth and bactericidal efficiency in Figure 3. It would be more convincing to show some experimental results that the blue light indeed helped phage penetration, or the phagocytosis may help the ROS generation induced by the blue light.
  4. For planktonic culture, the PBL+phage therapy showed the highest reduction in bacterial growth. Have the authors tried to use multiple dosing of the PBL or phage alone to test whether the reduction would be comparable to the combined method? It seems like phage is the major force for bacterial reduction, so is it possible that two doses of phage would achieve similar or even higher efficacy than the combined method?
  5. Another important thing that is missing is the reason why combining blue light pulse with phage method would be a better option than combining antibiotics with In terms of efficiency alone, the antibiotics+phage seems slightly better than PBL+phage according to the discussion section. The author should specify the innovation of the PBL+phage compared to current antibiotics+phage
Comments on the Quality of English Language

The English Language could be improved. For example, in the abstract part, "ANOVA/Kruskal-Wallis with post-hoc Tukey/Conover-Iman for comparisons (n=5 biological repeats over technical triplicates)." is not a complete sentence.

Round 2

Reviewer 1 Report

Comments and Suggestions for Authors

Authors have addressed all concerns raised in the previous report. I recommended accepting for publication. 

Comments on the Quality of English Language

Good. 

Reviewer 2 Report

Comments and Suggestions for Authors

Thank you for the response. The revised version addressed the previous concerns well.